# Good School Toolkit-Secondary Schools to prevent violence against students: protocol for a pilot cluster randomised controlled trial

Karen Devries ![ORCID],[1] Clare Tanton ![ORCID],[1] Louise Knight,[1] Janet Nakuti,[2] Barbrah Nanyunja,[2] Yvonne Laruni,[2] Mathew Amollo,[3] John Apota,[3] Timothy Opobo,[3] Jodie Pearlman ![ORCID],[1] Elizabeth Allen,[1] Chris Bonell ![ORCID],[4] Dipak Naker[2]

¹London School of Hygiene & Tropical Medicine, London, UK
²Raising Voices, Kampala, Uganda
³Africhild Centre, Makerere University, Kampala, Uganda
⁴Public Health and Policy, London School of Hygiene & Tropical Medicine, London, UK

**Correspondence to**
Dr Karen Devries;
karen.devries@lshtm.ac.uk

## ABSTRACT

**Introduction** No whole-school interventions which seek to reduce physical, sexual and emotional violence from peers, intimate partners and teachers have been trialled with adolescents. Here, we report a protocol for a pilot trial of the Good School Toolkit-Secondary Schools intervention, to be tested in Ugandan secondary schools. Our main objectives are to (1) refine the intervention, (2) to understand feasibility of delivery of the intervention and (3) to explore design parameters for a subsequent phase III trial.

**Methods and analysis** We will conduct a pilot cluster randomised controlled trial, with two arms and parallel assignment. Eight schools will be randomly selected from a stratified list of all eligible schools in Kampala and Wakiso Districts. We will conduct a baseline survey and endline survey 18 months after the baseline, with 960 adolescents and 200 teachers. Qualitative data and mixed methods process data collection will be conducted throughout the intervention. Proportion of staff and students reporting acceptability, understanding and implementing with fidelity will be tabulated at endline for intervention schools. Proportions of schools consenting to participation, randomisation and proportions of schools and individual participants completing the baseline and endline surveys will be described in a Consolidated Standards of Reporting Trials diagram.

**Ethics and dissemination** The ethical requirements of our project are complex. Full approvals have been received from the Mildmay Ethics Committee (0407-2019), the Uganda National Council for Science and Technology (SS 6020) and the London School of Hygiene & Tropical Medicine (16212). Results of this study will be published in peer-reviewed academic journals, and shared with public bodies, policy makers, study participants and the general public in Uganda.

**Trial registration number** PACTR202009826515511.

## STRENGTHS AND LIMITATIONS OF THIS STUDY

⇒ We are using mixed methods to refine the Good School Toolkit-Secondary Schools so that content is acceptable and delivery is feasible.
⇒ We select schools in a stratified manner to capture important sources of variation in implementation and parameters relevant to our phase III trial.
⇒ We rely on student reports of violence experience as our primary outcome, as student reports are likely to be biased in the opposite direction to the intervention effect.
⇒ Common to studies of behavioural interventions, it is not possible to blind participants to treatment allocation.
⇒ If we move forward to a phase III trial, this will represent one of the first interventions to address multiple forms of school violence to be trialled.

of violence in childhood is associated with increased risk of depression,[2] suicide,[3] sexual risk behaviours,[4] alcohol consumption,[5] adverse mental health outcomes, injury and disruptive behaviour,[6] poor employment outcomes[7] and poor performance on school tests.[8] Abused children are at increased risk for developing conduct disorder,[9] which predicts later use of violence in adult relationships.[10]

Despite the high prevalence and large health burden related to violence, few interventions are focused on reducing multiple forms of violence from different perpetrators in adolescence. Existing interventions to reduce violence in adolescence focus mainly on physical and emotional violence from peers,[11 12] and have overwhelmingly been conducted in the USA. Systematic reviews of interventions to prevent dating violence in adolescence generally find few effective programmes.[13 14] There are no systematic reviews of interventions to prevent violence

## INTRODUCTION

One billion children globally experience emotional, physical or sexual violence each year.[1] Longitudinal data show that experience

from school staff towards students,[15 16] despite findings that this form of violence is very common in some countries.[17] Other new interventions to prevent violence during adolescence have not been trialled in sub-Saharan Africa[18]; or focus mainly on intimate partner violence (IPV)[19] or sexual violence.[20]

Schools are a promising platform for delivery of prevention programming for adolescents. In particular, whole-school approaches to health promotion, which involve multiple actors within a school, including staff, students and administration, and are recommended by WHO as effective.[21] However, global systematic reviews of whole-school approaches to improving adolescent health find that interventions do not consider multiple forms of violence as outcomes.[22]

### Ugandan context

In Uganda, sexual violence and IPV are one of the top 12 leading causes of disease burden.[23] The 2015 national survey of Violence Against Children in Uganda found that 59% of young females and 68% of young males reported experience of physical violence, 34% of young females and 36% of young males reported emotional violence and 35% of females and 17% of males reported sexual violence under age 18 years.[24] Common perpetrators include teachers, peers and intimate friends, all of whom interact with students in school settings.[25] About 25% of adolescents attend at least some secondary school in Uganda.[26] There is a clear need for evidence-based interventions to reduce violence against adolescents in Uganda, and for whole-school approaches to violence prevention to be developed.

### Developing the Good School Toolkit-Secondary Schools

Raising Voices, a Ugandan non-governmental organisation (NGO), created the Good School Toolkit to reduce violence in primary schools (http://raisingvoices.org/good-school/). It is a whole-school approach which aims to change power dynamics underpinning violence, by changing the operational culture of the school. We tested the Toolkit in a cluster randomised controlled trial (RCT) of 42 primary schools in 2012–2014. The Toolkit reduced reported past week physical violence inflicted by school staff on young adolescents (mean age 13 years) by 42% (OR 0.40, 95% CI 0.26 to 0.64),[27] physical and emotional violence between peers and severe physical violence and injury from staff towards students.[28] The Toolkit is highlighted as an evidence-based strategy in the WHO INSPIRE guidelines for preventing violence against children.[29] Since 2015, Raising Voices has been adapting the Toolkit for use in secondary schools, creating a draft version of the Good School Toolkit-Secondary Schools (GST-S) in 2016. This paper describes the protocol for a pilot RCT of GST-S.

### Aims and research questions

The overall aim of this pilot trial is to determine whether progression to a phase III cluster RCT of the GST-S to prevent violence is justified. Our specific aims are: (1) to refine and finalise the GST-S intervention; (2) to understand feasibility of delivery of the intervention and (3) to explore design parameters for a subsequent phase III trial. Specific research questions are outlined in table 1.

### Trial partners

This pilot RCT is a collaboration between the London School of Hygiene & Tropical Medicine (LSHTM); Raising Voices and the AfriChild Centre, Makerere University.

## METHODS AND ANALYSIS
### Study design

The study is a pilot cluster RCT, with two arms and parallel assignment, whereby schools will be randomised to one of the two study arms to receive either the intervention or treatment as usual for the duration of the study. Surveys will be conducted twice, at baseline and 18 months afterwards (endline). Qualitative data and mixed-methods process data collection will be conducted throughout the intervention. The planned start date of the study was mid-2018, with baseline data collection at the end of 2018, and endline data collection 18 months after the start of implementation of GST-S. The actual dates of baseline and endline surveys were March 2020 and July 2023, respectively, due to multiple delays to the trial resulting from the COVID-19 pandemic and Ebola outbreak. These will be described fully in the trial results paper. The study was ongoing at the time of submission. Reporting of this protocol adhered to guidelines of the Standard Protocol Items: Recommendations for Interventional Trials 2013 checklist.

### Study area

The study will be conducted in Kampala and Wakiso Districts, Uganda.

### Selection of schools

Eight secondary schools will be included, selected randomly from a list of all eligible schools in Kampala and Wakiso Districts. Schools with >500 students, and with >50% day students will be eligible. Only schools which are mixed sex will be eligible, to ensure adequate numbers of participants reporting intimate and sexual relationships to allow us to explore material related to adolescent intimate relationships. Schools involved in other intervention studies will not be eligible. From eligible schools, we will consider stratifying based on factors likely to affect intervention acceptability and feasibility of delivery, including whether schools are: faith/non-faith, urban/rural; government/private. One pair and one reserve pair of schools will be selected from each stratum. If headteachers decline to participate, reserve schools will be invited.

### Allocation methods

An allocation sequence will be generated by EA. Headteachers will be invited to a meeting after the baseline

**Table 1** Aims and research questions

| Aims | Research questions |
|---|---|
| Refine intervention | 1. Is the intervention acceptable and understandable to male and female students of all ages, staff, caregivers and stakeholders? Is the GST-S coherent with the new activities? If not, how should the intervention be refined?<br>2. Do students/teachers/caregivers feel GST-S is meaningful to their lives? What kind of influence do activities have on participants, for example, do they foster reflection/create further dialogue? What key messages do participants take away with them after activities? |
| Feasibility of intervention delivery | 1. Is it feasible to implement the GST-S with fidelity over 18 months? Which activities do schools implement and why? What is the pace of implementation and intensity of different activities?<br>2. What are challenges and facilitators that relate to delivery of the intervention, and student, teachers and caregiver engagement with the intervention? How does the secondary school institutional context affect delivery of the intervention? What other similar programmes are being implemented at schools? |
| Estimate parameters for phase III trial | 1. For sample size calculations: What are preliminary estimates and 95% CIs for prevalence and clustering of phase III trial primary outcomes (sexual, physical and emotional violence from school staff, peers and intimate partners), adolescent intimate relationships and effectiveness of the intervention?<br>2. For measurement: Do proposed phase III outcome measures have construct validity and internal consistency? What is the best data collection mode (face-to-face interviews, computer-assisted self-completed surveys) and timing to minimise disruption in schools?<br>3. For recruitment and retention: Are secondary schools willing to be recruited and randomised, and will both intervention and control schools remain in a trial? What are individual student response rates? How many staff and student baseline and endline surveys can be linked, using name, age and identification numbers? What are levels of school attrition over time, particularly in control schools? What are participants and stakeholder views on caregiver consent? |

GST-S, Good School Toolkit-Secondary Schools.

survey and grouped into pairs of schools, matched on criteria described above. Each pair of headteachers will be given an opaque bag, into which they place their school names. A representative from each pair will then draw names out of the bag, with schools being allocated to intervention or control conditions according to the sequence, in the order they are drawn out of the bag. This is an unblinded study—following randomisation, schools will be aware of whether or not they are receiving the intervention or control condition.

## Intervention condition

Intervention schools will receive the GST-S, plus 'usual care'; currently in Uganda, there is no specific interventions or lessons related to violence prevention.

## The GST-S

The GST is a complex behavioural intervention that takes a whole-school approach to support students, staff, administration and caregivers through a series of steps designed to change the overall school culture. Implemented over 18 months, changes happen via 4 key entry points: (1) creating positive, mutually respectful relationships between teachers and students; (2) promoting student voice, participation, leadership and belonging and connectedness to school; (3) fostering more transparent and accountable school administrations and (4) strengthening engagement from caregivers and local leaders in the surrounding communities. It draws on the Transtheoretical Model by framing school-level change as a six-step process. It addresses power dynamics in relationships, which are thought to underpin multiple forms

of violence. There are about 60 different activities and core structures that schools can choose to implement, including those specifically designed to provide staff with alternatives to corporal punishment and foster critical reflection on power relations and violence. Materials include books describing activities, and booklets, posters, tools and example documents used to reinforce key ideas, all publicly available at www.raisingvoices.org. The GST is implemented by at least two teacher and two student 'protagonists' in each school, with mentorship and support from a trained violence prevention advocate from Raising Voices or another local child protection/child rights NGO.

We carried out a systematic process to adapt the GST-P for use in secondary schools.[30] This involved clarifying the logic model, conducting formative research (cross-sectional survey and focus groups) to improve understanding of the secondary school population, preparing new and modifying existing intervention components and pretesting new components with teachers and students. Based on the findings of this work and salient themes emerging in secondary schools, we strengthened existing content around gender and added new activities to address inequitable gender norms and promote gender equality. We also strengthened content around peer violence and added new activities to promote aspirational relationship values such as mutual respect and equal decision-making; critically reflect on power as it relates to sexuality and transactional relationships and address sexual violence in intimate relationships. This content is framed around relationships generally rather than IPV specifically, based on teachers' feedback in pretesting. We also strengthened materials around engaging with school administration, and created more opportunities for students to exercise leadership, reflecting their older age and increased agency.

In addition, other activities have been lightly adapted to increase effectiveness, dropped as they were difficult to implement and/or have had language changed to more appropriately address secondary school students. Content is described in table 2. Delivery of the GST-S will be led by two teacher protagonists and two student protagonists per school, with support from Raising Voices staff.

### Control group
Control schools will receive usual care plus the intervention after the endline survey.

### Sensitisation, recruitment and consent procedures
At national level, Raising Voices has presented the GST to relevant stakeholders at Ministry of Education and Sports; at the district level, meetings will be held with the District Education Officer in Wakiso and The Director Education and Social Services for Kampala to seek permission for the study to take place. For participants, we will employ a three-tiered strategy used successfully in our prior research[27]: written consent for school participation in the study and for school and classroom level assessments will

be sought from head teachers; parents of children in the school will be informed about the research in several ways and will be able to opt children out of being approached to participate in individual surveys and students below 18 years of age will be required to provide written informed assent, emancipated minors and students above 18 years of age will be required to provide voluntary written informed consent to participate in individual surveys. Children below the age of 18 years will be informed during the consent process that if interviewers feel their safety is at risk, they may have to discuss the case with the a Probation Officer.

### Outcomes
#### Primary outcomes
The main outcome of this pilot trial is whether criteria for progression to a phase III trial are met, based on criteria in tables 3 and 4 and judged against a traffic lights system.[27]

For the *acceptability* survey measure, we will develop a series of questions to assess whether specific activities and concepts are acceptable to teacher and student participants. Acceptability is a multidimensional concept,[31] we will focus the survey measure on: burden of participation and opportunity costs, perceived ethics and side effects, perceived effectiveness and general acceptability.

For the *understanding of content* survey measure, we will develop questions to gauge knowledge of concepts. Overall acceptability and understanding measures will be constructed as scores with different items weighted for importance.

For both acceptability and understanding, the proposed thresholds for progression have been defined based on expert opinion from Raising Voices' implementation team. These will be refined after survey questions are developed and piloted, but prior to endline data analysis. Briefly, we plan to convene a meeting to discuss scoring of items and finalise progression criteria thresholds, where expert implementers would feel comfortable with going ahead with widespread implementation (corresponding with 'green' progression criteria); where things could be implemented with some small refinements ('amber') and where scores indicated the Toolkit would need major refinements before implementation ('red').

*Fidelity* is defined as ≥50% of 21 new activities implemented and having core structures (student and staff protagonists, committees, court and school action plans) in place. This definition is based on Raising Voices' expert opinion and on the GST-P trial,[32] where all schools had core structures in place and reported implementing a mean of 19 activities over 18 months. Fidelity will be measured at the school level using data from structured observations. Based on the number of planned activities and structures successfully observed during the 32 randomly selected observations, we will extrapolate the total number of activities and modules implemented in each school. Fidelity will be further assessed via analysis of ongoing monitoring data collected as part of the GST-S

**Table 2**  Overview of the adapted GST-S content

| Step | School-led activities | Content of leadership workshop modules |
|---|---|---|
| **STEP 1—Creating Team/Network** | 1.1 Good school network<br>**1.2 Admin\* introduces GST to school**<br>**1.3 Recruit teachers to GSC**<br>**1.4 Recruit students to GSC**<br>**1.5 Recruit community members to GSC**<br>*1.6 Recruit admin\* to GSC*<br>**1.7 Subcommittee welcome meetings**<br>**1.8 Leadership workshop 1: GSC training (WS 1.1–1.8)**<br>*1.9 Good School Morning 1: Our Shared Rights (WS 1.6)* | **1.1 What is a Good School?**<br>1.2 Creating a conducive learning environment<br>1.3 What is a good teacher?<br>1.4 Creating positive discipline at your school<br>**1.5 What is good governance?**<br>*1.7 Four types of leaders*<br>**1.8 Using participatory facilitation** |
| **STEP 2—Preparing for Change** | 2.1 Create plan<br>**2.2 Survey**<br>*2.3 Bulletin board*<br>**2.4 Leadership workshop 2: GSC training (WS 2.1–2.7)**<br>**2.5 School-wide initiatives and activities**<br>*2.6 Good School Morning 2: Four Types of Leaders (WS 1.7)*<br>*2.7 One-week power campaign*<br>2.8 Launch GST | *2.1 Our shared rights*<br>**2.2 How are you using your power**<br>**2.3 Types of violence**<br>*2.4 Peer violence*<br>*2.5 Gender in schools*<br>**2.6 Challenging gender roles**<br>**2.7 Sexual violence in schools**<br>**2.8 Revisiting participatory facilitation** |
| **STEP 3—Good Teachers/Teaching** | 3.1 Create plan<br>**3.2 Leadership workshop 3: School staff (WS 3.1–3.8)**<br>**3.3 Student-teacher relationships**<br>**3.4 Creative teaching**<br>3.5 Bi-Monthly Teacher Meetings<br>**3.6 Professional goals & feedback**<br>*3.7 Good School Morning 3: Gender in Schools (WS 2.4)*<br>*3.8 Gender campaign* | **3.1 Remembering relationships**<br>3.2 Professional pride<br>*3.3 Challenging gender roles*<br>*3.4 Teaching for both genders*<br>**3.5 Creating teaching techniques**<br>3.6 Why do students misbehave?<br>3.7 Being a role model<br>*3.8 Why go to a Good School? (peer pressure)* |
| **STEP 4—Positive Discipline** | 4.1 Create plan<br>4.2 Leadership workshop 4: School staff (WS 4.1–4.7)<br>4.3 Reinforce positive discipline commitment<br>4.4 Recognise student strengths<br>4.5 Classroom rules<br>4.6 Student court<br>**4.7 School standards and rules**<br>*4.8 Good School Morning 4: Peer Violence (WS 2.3)*<br>*4.9 Peer violence campaign* | 4.1 What is corporal punishment?<br>4.2 Corporal punishment on trial<br>4.3 Punishment versus discipline<br>4.4 Why voice matters<br>4.5 Positive discipline responses<br>4.6 Positive discipline role-role play<br>4.7 Encouraging good behaviour |
| **STEP 5—Good Learning Environment** | 5.1 Create plan<br>**5.2 Create code of conduct**<br>5.3 Share code of conduct<br>5.4 Student leadership opportunities<br>*5.5 Prepare students for leadership (peer to peer)*<br>5.6 Create a student referral directory<br>5.7 Engage the community in caring for the physical compound<br>*5.8 Good School Morning 5: Sexual Violence in Schools (WS 2.6)*<br>*5.9 Good School parent's day* | 5.1 Smart choices<br>5.2 Address sexual violence in school<br>5.3 Team building and cooperation<br>5.4 Respect and responsibility<br>5.5 Self-esteem and value<br>5.6 Friendship and relationship<br>5.7 Gender and self-image |

Continued

**Table 2** Continued

| Step | School-led activities | Content of leadership workshop modules |
|---|---|---|
| STEP 6—School Governance/Way forward | 6.1 Create plan<br>*6.2 Good School Morning 6: Why go to a Good School? (WS 3.8)*<br>6.3 Good school assessment<br>6.4 Defining way the forward<br>6.5 Transition meeting<br>6.6 Community celebration | |

The original or unchanged GST content is in plain text, the strengthened content in bold text and the new content is in italics.
Admin, school administration; GSC, Good School Committee; GST-S, Good School Toolkit-Secondary Schools; WS, workshop.

intervention by Raising Voices staff and by school protagonists themselves. Fidelity will be further explored at the level of individual students and staff by adapting a 10-item measure from our GST-P trial to assess intervention exposure, reflecting activities in the GST-S. Records will be kept of schools approached and percentage of students responding to the baseline and endline survey will be calculated as a proportion of selected students consenting to and participating in the baseline and endline survey. We will therefore carefully describe what is implemented, when and why as well as what is not implemented, in order to produce a refined intervention with a high likelihood of compliance in a phase III trial.

## Other outcomes

Other outcomes include estimation of parameters in preparation for a phase III trial. To inform sample size for a phase III trial, we will explore the prevalence and clustering of phase III trial primary outcomes (including physical, sexual and emotional violence from school staff, peers and intimate partners), adolescent intimate relationships and the effectiveness of the intervention. We will also explore the construct validity and internal consistency of phase III intermediate and secondary outcomes (including gender attitudes, student participation, relationship quality, staff well-being and use of violence, resilience and mental health).[33 34] We will additionally explore the best data collection mode for a phase III trial, given that our adaptation research suggested that computer-assisted interviews can be used to gather data on violence from students.[30] In a subsample of 100 adolescents, we will test the use of audio-computer-assisted self-interviewing and compare this with data collected via face-to-face interviews to determine which has higher levels of disclosure, greater ease of completion and lower cost.

## Statistical analysis

*Primary outcomes*: proportion of staff and students reporting acceptability, understanding and implementing with fidelity will be tabulated at endline for intervention schools. Proportions of schools consenting to participation, randomisation and proportions of schools and individual participants completing the baseline and endline surveys will be described in a CONSORT diagram.

*Other outcomes*: descriptive summaries of baseline and follow-up data will be computed by arm. In accordance with CONSORT, no significance tests will be performed to test for differences at baseline, or given that this is a pilot trial, at follow-up. Descriptive statistics for continuous variables will include the mean, SD, median, range and number of observations. Categorical variables will be presented as numbers and percentages. Exploratory analysis for the phase III trial primary outcomes will be by intention-to-treat and include estimates of effect sizes with CIs and also estimates of the intracluster coefficient (ICC) for key outcomes. No interpretation will be made of any effect sizes or ICCs and findings will only be used to inform phase III trial design. Analyses will be carried out blind to treatment allocation.

**Table 3** Overall categories for progression

| Overall judgement | Criteria (with respect to outcomes in table 2) |
|---|---|
| Green—OK to progress | ► No more than one amber ratings for outcomes concerning 'intervention implementation' and 'research feasibility'<br>► No red ratings |
| Amber—progress with plans in place to address specific issues | ► No more than two amber ratings for outcomes concerning 'intervention implementation' and 'research feasibility'<br>► No more than one red rating |
| Red—no progress without further feasibility work | ► More than two amber ratings for 'research feasibility'<br>► More than one red rating |

**Table 4** Specific criteria for progression and their colour coding

| Type of outcome | Topic | Definition for categorisation of progression criteria | | |
| --- | --- | --- | --- | --- |
| | | Green | Amber | Red |
| Intervention implementation (intervention schools) | Acceptability | >80% staff* and students report acceptability in endline survey | ≥70% staff* and students report acceptability in endline survey | <70% staff* and students report acceptability in endline survey |
| | Understanding | >80% staff* and students report understanding in endline survey | ≥70% staff* and students report understanding in endline survey | <70% staff* and students report understanding in endline survey |
| | Fidelity | ≥3 of 4 intervention schools implement with fidelity at endline<br>► ≥50% of 21 new activities implemented and core structures in place | N/A | <3 of 4 intervention schools implement with fidelity at endline<br>► ≥50% of 21 new activities implemented and core structures in place |
| Research feasibility (all schools) | Enrolment | ≥6 of 8 schools agree to participate in the baseline survey | 5 of 8 schools agree to participate in the baseline survey | <5 of 8 schools agree to participate in the baseline survey |
| | Randomisation | ≥6 of 8 schools participating in the baseline survey agree to be randomised | 5 of 8 schools participating in the baseline survey agree to be randomised | <5 of 8 schools participating in the baseline survey agree to be randomised |
| | Follow-up | ≥6 of 8 schools randomised agree to endline data collection | 5 of 8 schools randomised agree to endline data collection | <5 of 8 schools randomised agree to endline data collection |
| | Response rate | >70% of eligible students respond to surveys | >60% of eligible students respond to survey | <60% of eligible students respond to survey |

*Staff are those involved with the intervention, that is, teacher protagonists, the teachers committee and the administration.
N/A, not available.

## Baseline and endline surveys

### Selection of students within schools

Through our discussion with headteachers prior to the baseline survey, we will determine whether it is feasible to collect data from a random sample of students within each school; if so, we will stratify by grade (senior 1–6) and sex to ensure equal numbers (selecting a total 960 students across all schools) from lists of all enrolled students in those classes. If not, we will sample whole lessons of students at baseline, stratified by grade. Where whole lessons of students are sampled, we will link baseline and endline surveys, to assess individual attrition and whether it would be possible to conduct a longitudinal survey in a phase III trial. Most students in secondary schools are aged between 13 and 18 years.

### Inclusion and exclusion criteria

Eligibility criteria for participation in research for children are: enrolled at participating schools, parents do not opt them out of the study and voluntary written informed consent/assent for participation from the child. Exclusion criteria are inability to speak or read either English or Luganda, an inability to provide informed consent or having a significant cognitive impairment that would prevent them from completing the questionnaire. Language ability of the children will be assessed by interviewers, and school staff will nominate children who may have a cognitive impairment that would prevent them from completing the questionnaire.

### Selection of staff within schools

Within each school, between 25 and 30 staff will be invited to participate in baseline and endline surveys. Eligibility criteria for staff will include employed at participating school and provides voluntary written informed consent to participate in the study.

### Data collection

All individual survey measures will be translated into Luganda, and back translated to English. Measures will be pilot tested at a school in Kampala to ensure that the items are understood by students and staff. Baseline questionnaires for students and staff have been structured and will be refined such that the average completion time for

students is approximately 1 hour and average time for staff is approximately 30 min. The final questionnaires will be pilot tested for overall length and flow, and the number of questions asked will be reduced such that overall completion time is as above. Endline surveys will be the same but include additional questions on acceptability of the Toolkit activities, understanding of the Toolkit, programme fidelity and student participation.

### Sample size

Since this is a pilot trial, no power calculations have been performed. Survey data will be collected from all school staff (n=approximately 200), and a sample of students (n=approximately 960 total across all schools). These numbers have been determined by a combination of considerations, including how many schools Raising Voices was able to implement GST-S in and school stratification characteristics the implementation team felt were important to vary in testing (see Selection of schools), and to give a large enough sample size to explore differences in measure suitability by age and grade of students. In a phase III trial, preliminary sample size calculations suggest that for at least 80% power and 5% significance, we will require 25 schools per arm to detect a relative reduction of 33.3% in IPV against girls (absolute reduction 5%). This assumes 100 girls per school at endline; drop out of two schools per arm; ICC of 0.03 and prevalence of IPV against girls of 15% among those in the control arm at follow-up. These calculations will be refined with data from this phase II trial.

### Qualitative and process data collected in intervention schools
#### Structured observations of intervention activities

A total of 64 observations (4 intervention activities×4 schools×4 terms) will be carried out by the research team in intervention schools. Observations will be on a mix of: (1) planned activities purposively selected by Raising Voices, to inform refinement and (2) at times randomly selected by the research team, to document what elements of the GST-S are in place, activities occurring and student and staff uptake of ideas.

#### Monitoring data from GST-S intervention

Monitoring data will be routinely collected as part of the GST-S intervention, by Raising Voices staff and by the school protagonists; including the number and types of support visits to schools; the number of activities planned and implemented each term; number of attendees at each activity (staff, students, caregivers, age and sex), time spent in preparation, impressions of understanding, successes and challenges.

#### Focus group discussions

Focus group discussions (FGDs) with ordinary and advanced level classes, male and female students (4 groups×2 schools×3 times=24 FGDs), with male and female staff, caregivers, Raising Voices staff (4 groups×3 times=12 FGDs) will be conducted twice during implementation and once after implementation, to understand

progress, challenges and suggestions for refinement. For the student FGDs, one urban and one rural school will be randomly sampled out of the for intervention schools. Student and staff participants will be purposively selected initially according to have varying involvement and engagement with the intervention activities, but we will also be guided by other factors emerging as important.

#### Qualitative in-depth interviews

In-depth interviews (IDIs) (total=64) will be conducted with approximately 8 stakeholders, 6 Raising Voices staff, 10 school staff and 40 students across sex and age groups. Experiences of intervention activities and suggestions for refinement will be explored. There will be flexibility in the numbers, timing of interviews and sampling strategy. Similar to the approach used for FGDs, student and staff participants will be purposively selected to have varying involvement with intervention activities, but we will also be guided by other factors emerging as important.

#### Structured interviews with headteachers

A structured interview will be conducted with headteachers at the beginning and end of implementation in both intervention and control schools to document basic information about the school including numbers of students enrolled and progression; numbers of staff; details of other health promotion activities happening, especially around sexual and reproductive health education and contamination and to describe any existing violence prevention programming.

#### Interview procedures

Interviewers will receive three full weeks' training including instruction on how to collect data in private so that disclosures from staff and students cannot be overheard, how to stop the interview if participants become distressed and how to refer participants for support. Interviews will take place where they cannot be overheard and the interviewer will confirm that the participant is happy proceeding at this time and in the location before commencing the interview. If the situation changes, people move nearby or the interviewee appears uncomfortable or uneasy, then the interviewer will check in, stop the interview and move to another location, or time, as necessary. Participants will be informed during the informed consent procedure, and at the end of the interview, that if they feel they would like to talk to a counsellor about something that has come up during the interview then we can refer them to talk to someone at local services, as detailed below under referral procedures. Reasons for non-consent will be recorded by the interviewer.

#### Data management

Identifying information will be captured separately from questionnaire responses. Students and staff will be identified initially using lists provided by the headteacher. An electronic dataset containing names and ID numbers will be created and stored in a password-protected file on a secure server at LSHTM and on a repository desktop

computer at the AfriChild Centre. This computer is only accessible to the IT manager and research manager. All procedures will be compliant with General Data Protection Regulation.

## Ethics and dissemination
### Ethical approvals
Ethical approval for the study has been obtained from the LSHTM Ethics Committee, the Mildmay Uganda Research Ethics Committee and the Uganda National Council for Science and Technology. Adverse and serious adverse events and reactions will be reported to LSHTM as the study sponsor, and each ethics committee. Serious events and reactions will be reported within 24 hours.

### Child protection and safeguarding
Participants will be asked about experience of violence which may be severe, and in some cases, they may be at immediate risk of further violence and/or other acute health difficulties. Adolescents will be referred onwards to local child protection partners according to the type, severity and timeframe of violence exposure disclosed. This process will be overseen by a named and contracted partner from a local child protection organisation. This partner organisation will receive all referrals from the study team and will take further actions in accordance with their professional opinion and local best practice.

In brief, all children who participate in the study, regardless of what they disclose, will be offered the opportunity to visit a trained counsellor who is fluent in Luganda. Children who disclose mild to moderate experiences of violence will be offered referral to a counsellor. For children who disclose more severe experiences of violence in the past week or past year, the child protection officer from the independent child protection organisation will be informed, and will refer cases onwards in accordance with local policy. Children who disclose recent sexual violence or severe physical violence or injury, or who feel suicidal, will be taken immediately to a health centre, and the child protection officer from the independent child protection organisation and District Probation officer will be informed so that further follow-up may take place.

Referrals will not be mandatory (adolescents can choose not to take them up, except where required by law for very severe incidents of violence or where the child is perceived to be in danger) but will be heavily facilitated (adolescents will be transported if they would like, followed-up and encouraged to attend, while respecting their wishes and evolving autonomy). In Uganda, 'defilement', or rape of children is illegal, and will be referred by the independent child protection organisation to District Probation officers and to the police. A small team will support the independent child protection organisation to ensure responses to cases are appropriate and timely, as in our other research in this setting.[35]

### Adult referrals
Teachers are also being asked about their experiences of violence and current mental health. Any adult who has experienced violence or need mental health support, we will provide a list of local organisations who can be contacted and provide appropriate support. This will be optional for adults. For any adult who discloses use of severe physical violence in the past year or report having ever had sex with a student, the partner organisation will be informed and cases will then be handled as appropriate, as relevant authorities including the police, District Education Officer and others may be informed.

### Dissemination
Partnership with AfriChild will ensure that we benefit from local academic expertise and that we contribute to capacity building in trial methodology. Raising Voices already has experience of delivering the GST-P in >750 Ugandan primary schools in partnership with local NGOs. Raising Voices is currently discussing collaboration with the Ministry of Education and Sports (MoES) to provide the GST-P to all Ugandan primary schools . If a phase III trial shows effectiveness, there is a high likelihood that the intervention can be widely implemented.

### Patient and public involvement
Raising Voices has maintained one advisory group including student users, teachers, school staff and MoES officials, over the lifetime of the primary and secondary editions of the Toolkit. For previous work around the adaptation of the Toolkit, we convened new adolescent and teacher steering groups that have provided feedback on what types of violence are of concern in secondary schools, adolescent relationships and ideas about intervention activities. We will maintain collaboration with all three of the advisory groups throughout the project. We expect the teacher and adolescent groups to play a major role in collaborating on the refinement of the content and delivery mechanisms, and the larger advisory group which includes MoES staff to continue to advise on how best to embed the GST-S within secondary schools. We will meet with advisory groups at the project inception, during pretesting of measures and recruitment, and again after our data are collected to share and jointly interpret results.

## DISCUSSION
The GST-S is a whole-school intervention that aims to address multiple common forms of violence during adolescence, and is designed to be implemented at low cost in resource-poor settings. An eventual phase III trial of the GST-S will represent the first test of a whole-school approach to prevention of multiple forms of violence in adolescence in sub-Saharan Africa.

Whole-school prevention approaches are recommended by WHO and are effective for a range of health conditions.[21] School-based approaches can potentially

reach a wide cross-section of adolescents, however, they may also compete for class time and rely on school personnel who lack training. Thus, trialling approaches to quantify benefits and harms is important.

## Strengths and limitations

Common to studies of behavioural interventions, it is not possible to blind participants to treatment allocation, which may lead to overestimates of treatment effect in a phase III trial. Outcomes are self-reported, which is considered gold-standard in violence research as third party reports vastly underestimate prevalence,[36] particularly for more stigmatised forms of violence. However, participants in violence prevention programmes may under-report their use of violence as a result of intervention exposure, due to both socially desirable responding and response shift bias (where they become more aware of concepts of violence and shift understanding, and reporting, of their own experience[37]). For this reason, we rely on student reports of violence experience as our primary outcome, as student reports are likely to be biased in the opposite direction to the intervention effect.

This is a pilot trial, hence we are working in a small number of schools. Although we have selected these in a stratified manner to capture important sources of variation in implementation and parameters relevant to our phase III trial, it is possible that there are other school-level characteristics which will affect intervention implementation that we have not been able to explore here.

## Implications

The results of this phase II trial will assist others planning phase III trials on similar topics, by providing information on content acceptability and understanding, the process of refining the intervention to increase acceptability and understanding and basic information on design parameters including clustering of relevant outcomes. Many of the issues facing Ugandan secondary school students are common elsewhere in sub-Saharan Africa, thus our results will be broadly informative in other settings.

**Contributors** KD designed the study, obtained funding and drafted the manuscript. CT manages all aspects of the study, input into the design of the study and co-drafted the manuscript. LK, EA and CB input into the design of the study and critically revised the manuscript. JN, BN and YL contributed to the design, implementation and refinement of the intervention, input into the study design and critically revised the manuscript. MA, JA and TO participated in data collection, input into the design of the study and critically revised the manuscript. JP read and critically commented on the manuscript. DN designed the intervention, input into the design of the study, obtained funding and critically revised the manuscript.

**Funding** This research is jointly funded by the UK Medical Research Council (MRC) and the Foreign Commonwealth and Development Office (FCDO) under the MRC/FCDO Concordat agreement (MR/R022208/1).

**Competing interests** DN, JN, BN and YL are employed by Raising Voices, and are all involved in the design and implementation of the Good School Toolkit-Secondary Schools.

**Patient and public involvement** Patients and/or the public were involved in the design, or conduct, or reporting, or dissemination plans of this research. Refer to the 'Methods' section for further details.

**Patient consent for publication** Not required.

**Provenance and peer review** Not commissioned; externally peer reviewed.

**ORCID iDs**
Karen Devries http://orcid.org/0000-0001-8935-2181
Clare Tanton http://orcid.org/0000-0002-4612-1858
Jodie Pearlman http://orcid.org/0000-0002-7651-2067
Chris Bonell http://orcid.org/0000-0002-6253-6498

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
