## [Reviewer comments · BMJ Open]

ARTICLE DETAILS

TITLE (PROVISIONAL)	The Good School Toolkit - Secondary Schools to prevent violence against students: Protocol for a pilot cluster randomised controlled trial
AUTHORS	Devries, Karen; Tanton, Clare; Knight, Louise; Nakuti, Janet; Nanyunja, Barbrah; Laruni, Yvonne; Amollo, Mathew; Apota, John; Opobo, Timothy; Pearlman, Jodie; Allen, Elizabeth; Bonell, Chris; Naker, Dipak

VERSION 1 – REVIEW

REVIEWER	Richards, David University of Exeter, Institute for Health Research
REVIEW RETURNED	26-Oct-2023

GENERAL COMMENTS	I will confine my review to the feasibility methods as I have no expertise in the intervention. Ethics: although the team have received ethics approvals and have a commendable section on safeguarding, some more details on their ethical principles is required Outcome definitions: very clear progression criteria are listed, graded in a traffic light system. However, there needs to be a clearer justification for the selection of these criteria and progression metrics specifically. For example why is >80% acceptability green and <70% red? Reference to previous studies or some other justification would help here. Secondly, I could not find a list of the outcomes to be collected to inform the phase three sample size collection (at least it was not clear to me in the section on outcomes). I totally agree that the focus is on feasibility of intervention and methods, but these outcomes are mentioned in table 1 section 3 and yet not clearly outlined in the text. It does not distract from the feasibility objectives to have these listed provided the analysis section remains clear at the lack of between group estimating. Sample size: the authors correctly state that a sample size calculation based on between group outcome differences is not appropriate for this type of study. Nonetheless, they are clear about both cluster and individual sample sizes, but there is no justification for these. Why are the (quite large) sample participant numbers and cluster numbers chosen? How can they and we be sure that these numbers are sufficient (or alternatively too large) to answer their feasibility questions? When answering this, please do not use the circular logic against the criteria expressed in the progression criteria, but find some further (maybe past literature in
--

	this area) to clarify why eight clusters and nearly 1000 participants have been targeted. The reference 30: "Grundlingh, H. and K.M. Devries, GST-S adaptation paper. under review." needs a clearer reference and maybe some detail in the text on the methods used to adapt the intervention to the current context and the results of these efforts.
--	---

REVIEWER	Mahlangu , Pinky South African Medical Research Council
REVIEW RETURNED	02-Nov-2023

GENERAL COMMENTS	Thank you for the opportunity to review this robust work, a well-conceptualized cluster pilot RCT which aims 1) to refine and finalise the GST-S intervention; 2) to understand the feasibility of delivery of the intervention; and 3) to explore design parameters for a subsequent phase 3 trial through mixed methods which will include baseline and endline survey at 18 months after baseline with 960 adolescents and 200 teachers in 8 schools, and includes a range of qualitative and process evaluation throughout the 18 months period. Below are some reflections and clarity seeking questions to help strengthen the methods.  • The study is a pair-matched pilot cluster RCT, with two arms and parallel assignment. A few lines explaining what is meant by a parallel assignment will benefit readers who are not quantitative researchers. • Delivery of the GST-S will be led by two teacher protagonists and two student protagonists per school, what are the implementers going to receive to compensate for their time during the long- term implementation of the intervention • The protocol states that students below 18 years of age will be required to provide written informed assent. Are their parents going to be asked to provide written consent, other than being able to opt out children for participation in surveys, please clarify • Acceptability survey measure needs to be expanded to clearly show how the researchers will measure acceptability of content of the intervention 8 schools will be randomly selected. Measuring acceptability of content should include whether the intervention addresses the unique needs/challenges of teachers and students in secondary schools. It is also not clear to the reader which specific activities and concepts will be measured in the content of the intervention, and which will be excluded. • It is not clear how you are going to ensure that N for school staff is approximately 200, if within each school between 25 and 30 staff will be invited to participate in baseline and endline surveys, and you only have 8 schools to participate in the pilot • The protocol indicates that focus group discussions will be conducted: 4 groups x 2 schools x 3 times, but not clear which of the schools are the 2 schools, and how they will be selected from the 8 participating schools? Who (which staff and learners) will you invite to participate in FGDs, and what criteria will be used to purposively select learners and staff for FGDs • Elaborate on the criteria you will use to select participants for the 64 IDIs • Child protection and safeguarding listed in the protocol is well appreciated. However, the ethical considerations on the protocol only focuses on the child, and there is omission for ethical considerations for school staff who will also participate in the pilot. Researchers are encouraged to include ethical considerations for staff participants
--

	 • It is not clear in the protocol how the data collected through various methods will be used to refine the intervention and ultimately achieve aim 1 of the study. Clarification of how and when that step will be conducted will help enhance the methodology.
--	--

VERSION 1 – AUTHOR RESPONSE

Reviewer: 1

Prof. David Richards, University of Exeter

Comments to the Author:

I will confine my review to the feasibility methods as I have no expertise in the intervention.

Ethics: although the team have received ethics approvals and have a commendable section on safeguarding, some more details on their ethical principles is required

***Thank you for this comment, however we are unclear on what specific details the reviewer might be referring to. If this can be clarified we are happy to add more detail.**

Outcome definitions: very clear progression criteria are listed, graded in a traffic light system. However, there needs to be a clearer justification for the selection of these criteria and progression metrics specifically. For example why is >80% acceptability green and <70% red? Reference to previous studies or some other justification would help here.

***Thank you for this very helpful comment. We have added detail to clarify that these thresholds are determined via discussion with Raising Voices’ expert implementation team, based on their assessment of what constitutes ‘green’, ‘amber’ and ‘red’ levels of acceptability and understanding. We recognise that this is subjective, but there are no standardised questions available to determine acceptability (as these need to be intervention specific), and there is no standardised threshold of ‘acceptability’ associated with the traffic light system (as this would also be unique to each intervention). Implementation experts are best placed to set these thresholds, and thus we have described our process for formalising this input in the text.**

Secondly, I could not find a list of the outcomes to be collected to inform the phase three sample size collection (at least it was not clear to me in the section on outcomes). I totally agree that the focus is on feasibility of intervention and methods, but these outcomes are mentioned in table 1 section 3 and yet not clearly outlined in the text. It does not distract from the feasibility objectives to have these listed provided the analysis section remains clear at the lack of between group estimating.

***Thank you for raising this. We have added further details of outcomes related to our phase 3 trial to ensure that are outcomes are aligned with the aims included in Table 1.**

Sample size: the authors correctly state that a sample size calculation based on between group outcome differences is not appropriate for this type of study. Nonetheless, they are clear about both cluster and individual sample sizes, but there is no justification for these. Why are the (quite large) sample participant numbers and cluster numbers chosen? How can they and we be sure that these numbers are sufficient (or alternatively too large) to answer their feasibility questions? When answering this, please do not use the circular logic against the criteria expressed in the progression criteria, but find some further (maybe past literature in this area) to clarify why eight clusters and nearly 1000 participants have been targeted.

***We have added text to describe how sample size decisions were made. “These numbers have been determined by a combination of considerations, including how many schools Raising**

Voices was able to implement GST-S in and school stratification characteristics the implementation team felt were important to vary in testing (see school selection criteria), and to give a large enough sample size to explore differences in measure suitability by age and grade of students.”

The reference 30: "Grundlingh, H. and K.M. Devries, GST-S adaptation paper. under review." needs a clearer reference and maybe some detail in the text on the methods used to adapt the intervention to the current context and the results of these efforts.

***We have revised this reference and added detail around the adaptation process where this reference is cited.**

Reviewer: 2

Dr. Pinky Mahlangu , South African Medical Research Council

Comments to the Author:

Thank you for the opportunity to review this robust work, a well-conceptualized cluster pilot RCT which aims 1) to refine and finalise the GST-S intervention; 2) to understand the feasibility of delivery of the intervention; and 3) to explore design parameters for a subsequent phase 3 trial through mixed methods which will include baseline and endline survey at 18 months after baseline with 960 adolescents and 200 teachers in 8 schools, and includes a range of qualitative and process evaluation throughout the 18 months period. Below are some reflections and clarity seeking questions to help strengthen the methods.

- The study is a pair-matched pilot cluster RCT, with two arms and parallel assignment. A few lines explaining what is meant by a parallel assignment will benefit readers who are not quantitative researchers.

***Thank you for your suggestion. We have added in additional detail under Study Design to further explain the concept of parallel assignment.**

- Delivery of the GST-S will be led by two teacher protagonists and two student protagonists per school, what are the implementers going to receive to compensate for their time during the long- term implementation of the intervention

***Thank you for raising this important point. As per the GST-S intervention protocols, teacher and student protagonists are not paid with monetary compensation for their time. Instead they participate in the intervention voluntarily. This is considered by Raising Voices to be an important aspect of ensuring the intervention is sustainable and can be implemented in low-resource settings.**

- The protocol states that students below 18 years of age will be required to provide written informed assent. Are their parents going to be asked to provide written consent, other than being able to opt out children for participation in surveys, please clarify

***Thank you for raising this query. All of our consent procedures were approved by ethical review committees in both Uganda and the UK. We did not ask caregivers to provide written consent, as that would have unduly prevented vulnerable children from participating. Instead, any concerned caregiver could only opt children out of being approached to participate. Headteachers were instead asked to provide consent on behalf of students, which is common in other school-based survey research on violence in Uganda, and children themselves also then provided consent. The three-tiered strategy that is detailed in the protocol was found to be acceptable among headteachers, staff, students and caregivers in the previous trial of the Good School Toolkit Primary Schools trial. We have detailed the approach under “Sensitisation, recruitment and consent procedures”, and also referenced the primary schools**

trial which used the same approach.

- Acceptability survey measure needs to be expanded to clearly show how the researchers will measure acceptability of content of the intervention 8 schools will be randomly selected. Measuring acceptability of content should include whether the intervention addresses the unique needs/challenges of teachers and students in secondary schools. It is also not clear to the reader which specific activities and concepts will be measured in the content of the intervention, and which will be excluded.

***Thank you for raising this, and we agree that measurement of acceptability is a very important area. Unfortunately there is no pre-existing, standard approach we can use to measure acceptability, as it is highly linked to the intervention content, which is new. We therefore need to develop and pilot survey questions related to acceptability. We have added more detail about this in response to comments from Reviewer 1, in the methods section of the text, clarifying that thresholds will be set before the endline data analysis.**

- It is not clear how you are going to ensure that N for school staff is approximately 200, if within each school between 25 and 30 staff will be invited to participate in baseline and endline surveys, and you only have 8 schools to participate in the pilot

***Thank you for pointing this out. We will invite around 30 staff per school, so that there is a slight oversample to bring our total sample to approximately 200. We also note that the staff survey is not our primary objective in this pilot study; rather we are taking the opportunity to pilot a staff questionnaire (so if the sample is slightly under 200 this is acceptable).**

- The protocol indicates that focus group discussions will be conducted: 4 groups x 2 schools x 3 times, but not clear which of the schools are the 2 schools, and how they will be selected from the 8 participating schools? Who (which staff and learners) will you invite to participate in FGDs, and what criteria will be used to purposively select learners and staff for FGDs

***Thank you for your comment. For the student focus group discussions, one urban and one rural school will be randomly sampled. Our initial plan is to also sample according to involvement/engagement with the intervention activities but we will also be guided by other factors emerging as important. We have added this additional detail to the protocol.**

- Elaborate on the criteria you will use to select participants for the 64 IDIs

***Similar to our approach used for FGDs, we will purposively sample according to varying involvement with the intervention, but also be guided by other emerging factors. We have also added this detail into the manuscript.**

- Child protection and safeguarding listed in the protocol is well appreciated. However, the ethical considerations on the protocol only focuses on the child, and there is omission for ethical considerations for school staff who will also participate in the pilot. Researchers are encouraged to include ethical considerations for staff participants

***Thank you for raising this important point. We have a procedure in place for school staff who report experience or use of violence, or who need mental health support. We have detailed the procedure within the Ethics section of the manuscript.**

- It is not clear in the protocol how the data collected through various methods will be used to refine the intervention and ultimately achieve aim 1 of the study. Clarification of how and when that step will be conducted will help enhance the methodology.

***Thank you for your comment. Several methods will contribute to refining the intervention. We will explore fidelity at the school and individual levels of staff and students from the endline survey, routine monitoring data and structured observations. This will therefore describe what is implemented, when and why, as well as what is not implemented, in order to produce a refined intervention with a high likelihood of compliance in a phase 3 trial. Our qualitative work will further contribute to refinement of the intervention as FGDs and IDIs may highlight suggestions for refinement of activities and content. Raising Voices and engagement with local groups will play a key role in collaborating on the refinement of content and delivery based on the findings that emerge, as details in the 'Involvement of local groups' section. We have added additional detail within the description of the Fidelity outcome to reinforce the link between measuring fidelity and refining the intervention.**